# Experimental QND measurements of complementarity on two-qubit states with IonQ and IBM Q quantum computers

Nicolas Schwaller,[1, 2, ∗] Valeria Vento,[1, †] and Christophe Galland[1, ‡]

[1]*Institute of Physics, École Polytechnique Fédérale de Lausanne (EPFL), Lausanne, CH-1015, Switzerland*
[2]*Miraex, EPFL Innovation Park, Bâtiment L, Lausanne, CH-1015, Switzerland*
(Dated: July 21, 2021)

We report the experimental nondemolition measurement of coherence, predictability and concurrence on a system of two qubits. The quantum circuits proposed by De Melo *et al.* [1] are implemented on IBM Q (superconducting circuit) and IonQ (trapped ion) quantum computers. Three criteria are used to compare the performance of the different machines on this task: measurement accuracy, nondemolition of the observable, and quantum state preparation. We find that the IonQ quantum computer provides constant state fidelity through the nondemolition process, outperforming IBM Q systems on which the fidelity consequently drops after the measurement. Our study compares the current performance of these two technologies at different stages of the nondemolition measurement of bipartite complementarity.

## INTRODUCTION

Interest in quantum nondemolition (QND) measurements dates back to the early ages of quantum theory [2]. In the 1930s, they were proposed to overcome the limitations imposed by Heisenberg's uncertainty principle, via repeated measurements of a quantum state. In quantum mechanics, the measurement process does produce a perturbation (also called "back-action") on the state of the measured system. In general, back-action limits the precision of a measurement by increasing the standard deviation of the measured observable upon repeated measurements. However, a QND measurement on a system allows to measure an observable without back-action on this observable, even if the state of the system itself is affected by the measurement.

Three criteria are commonly used to assess the quality of a QND measurement [3, 4]. The first one is the *quantum state preparation* criterion. An ideal QND measurement of a given observable projects the conditional quantum state on an eigenstate of this observable, and thus can be purposed to state preparation. Second, the *measurement accuracy*: the QND measurement should give the expected outcome for a given input state. The last one is the *nondemolition* criterion, which states that the observable should not be disturbed by the measurement. If the input state is an eigenstate of the measured observable, its evolution through the QND measurement should be given by the identity operator. It can be evaluated by recording the difference between measurement outcomes on the observable before and after the QND measurement. Most QND implementations are related to an observable measured on a single-partite quantum system, e.g. the spin readout of an electron [5] or the position readout on a mechanical oscillator [6, 7].

Multipartite systems, on the other hand, can sustain internal quantum correlations such as entanglement, and their manipulation represents an essential resource for

quantum technologies. For the simplest nontrivial multipartite quantum system consisting of two qubits, labeled $A$ and $B$, Jakob and Bergou have shown that a complementarity relation holds between bipartite (nonlocal) and single-partite (local) properties [8], which has recently been checked experimentally on a truly bipartite system [9], with IBM Q. The Jakob-Bergou relation contains three quantities: first, the *concurrence* $\mathscr{C}$, an entanglement measure that is genuinely bipartite [10]. Second, the *coherence* or *visibility* $\mathscr{V}_A, \mathscr{V}_B$ of each qubit and, third, their respective *predictability* $\mathscr{P}_A, \mathscr{P}_B$, which are all single-partite, local properties defined for each subsystem. The "triality" relation $\mathscr{C} + \mathscr{V}_k + \mathscr{P}_k = 1$ ($k = A, B$) generalizes the wave-particle duality to bipartite systems, in the same manner as coherence and predictability express the wave-particle duality of a single qubit [1]. Moreover, the projective measurement of one of these observables induces a maximal uncertainty on the two other complementary observables.

These fundamental quantities that characterize qubit states are the subject of active experimental research due to their relevance for quantum computing, as they are at the heart of the essential concepts of entanglement and interference, see e.g. [11, 12]. If the recent availability of circuit quantum computers opens an opportunity to test predictions on physical qubits, the limits of these noisy, intermediate scale machines are quickly reached. Various methods aiming at the measurement of these quantities exist, which have their own sensitivity to quantum decoherence. In [9], linear state tomography was used on the qubits of IBM Q to minimize the circuit size and length, hence the decoherence-induced noise. Circuits implementing QND measurements of the same complementary quantities were proposed in 2007 by De Melo *et al.* [1]. These circuits are restricted to quantum states with real coefficients, which does not affect their relevance for quantum information processing [13]. As for all QND circuits the outgoing state can be used as a resource for further processing, since postselection via the

measurement of an ancilla system ensures the projection on an eigenstate of the measured observable. In particular, measurement of the concurrence can be used for heralding pure Bell states, acting as a form of entanglement distillation. If efficient Bell state preparation is more than a decade old [14], their generation on quantum computers is an ongoing and relevant task [15]. We note that similar circuits are used by surface codes for error correction, where the QND outcome is used to detect errors and actively correct the state encoding information [16, 17], which is a key challenge to the improvement of *noisy intermediate-scale quantum* (NISQ) computers. Even though error rates of current quantum computers such as IBMQ are still too high for successful implementation of error correction, the prospect of using QND measurements in this task motivates a careful analysis of their performance on existing platforms.

The goal of the present paper is to thoroughly test the QND circuits of ref. [1] on two different platforms for the first time. To this end we make use of the newest IonQ trapped ion quantum computer [18] which feature qubits with extended coherence times compared to IBM Q. We shall compare the quality of the results on IonQ and on four superconducting qubit platforms of IBM Q [19]. Both systems are NISQ computers, but rely on different technologies, different connectivities and numbers of qubits. For each implementation we assess the three criteria of a good QND measurement, namely its efficiency for quantum state preparation, its measurement fidelity and its non-demolition character. Moreover, the actual benefits of the circuit optimization algorithms offered on IBM Q via *Qiskit* [20] are investigated, with outcomes highlighting the fragility of the optimization routine in the presence of slow circuit parameter fluctuations. Overall, we find that IonQ produces results closer to the ideal quantum model, most likely due to its lower gate error rate and optimized connectivity with respect to IBM Q. Our study provides an assessment of distinct quantum computing architectures based on the two technologies of today's most sophisticated quantum computers, namely superconducting circuits and trapped ions, in the fundamentally relevant task of QND measurement of complementary two-qubit observables.

## QND MEASUREMENT OF TWO-QUBIT COMPLEMENTARITY

In this section we present our state preparation circuit, which generates the input state to be injected in the QND measurement circuits. A brief summary of De Melo's proposal [1] and its relevant operational properties is also presented.

### Input state generation

To generate the input state $|\chi\rangle$, we use the three parameter circuit depicted in fig. 1. The resulting state is

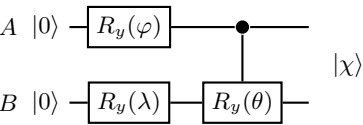

Fig. 1: Quantum circuit generating the input state $|\chi\rangle$. $\varphi, \theta, \lambda$ are real parameters in $[0; 2\pi]$. $R_y(\varphi)$ and $R_y(\lambda)$ are single qubit rotations around the $y$ axis of the Bloch sphere, while $R_y(\theta)$ is a rotation of qubit $B$ around $y$, controlled by the state of qubit $A$ in the computational basis ($z$ axis of the Bloch sphere).

written in the Bell basis as

$$|\chi\rangle = \alpha |\Psi^-\rangle + \beta |\Psi^+\rangle + \gamma |\Phi^-\rangle + \eta |\Phi^+\rangle \quad (1)$$

where the Bell states are defined as $|\Psi^\pm\rangle = \frac{1}{\sqrt{2}}(|10\rangle \pm |01\rangle)$ and $|\Phi^\pm\rangle = \frac{1}{\sqrt{2}}(|11\rangle \pm |00\rangle)$ for consistency with [1]. The circuit of fig. 1 generates $|\chi\rangle$ with real coefficients

$$\alpha = \frac{1}{\sqrt{2}} \left( \cos\frac{\lambda}{2}\cos\frac{\theta}{2}\sin\frac{\varphi}{2} - \sin\frac{\lambda}{2}\left[\cos\frac{\varphi}{2} + \sin\frac{\varphi}{2}\sin\frac{\theta}{2}\right] \right), \quad (2)$$

$$\beta = \frac{1}{\sqrt{2}} \left( \cos\frac{\lambda}{2}\cos\frac{\theta}{2}\sin\frac{\varphi}{2} + \sin\frac{\lambda}{2}\left[\cos\frac{\varphi}{2} - \sin\frac{\varphi}{2}\sin\frac{\theta}{2}\right] \right), \quad (3)$$

$$\gamma = \frac{1}{\sqrt{2}} \left( -\cos\frac{\varphi}{2}\cos\frac{\lambda}{2} + \sin\frac{\varphi}{2}\sin\left(\frac{\theta}{2} + \frac{\lambda}{2}\right) \right), \quad (4)$$

$$\eta = \frac{1}{\sqrt{2}} \left( \cos\frac{\varphi}{2}\cos\frac{\lambda}{2} + \sin\frac{\varphi}{2}\sin\left(\frac{\theta}{2} + \frac{\lambda}{2}\right) \right). \quad (5)$$

One can check that the coefficients of the superposition of Bell states (1) can all be continuously varied from 0 to 1 thanks to the parameters $\varphi, \theta, \lambda$. The full range of $\mathscr{V}_A, \mathscr{V}_B, \mathscr{P}_A, \mathscr{P}_B$ and $\mathscr{C}$ can be investigated by varying only the 2 parameters $\varphi$ and $\theta$ [9]. In fact, only the local quantities of the qubit $B$, namely $\mathscr{V}_B$ and $\mathscr{P}_B$, depend on $\lambda$, which is expected as $\lambda$ defines a local operation applied on qubit $B$. Finally, we note that any fixed value of $\lambda$ is compatible with the full range analysis, thus we will set $\lambda = 0$ for the experiment.

Let $\rho$ be the density matrix of a two-qubit ($A$ and $B$) state, and the reduced density matrices of subsystems $\rho_A = \text{Tr}_B(\rho)$, $\rho_B = \text{Tr}_A(\rho)$. In this work, the definitions of eqs. (6) to (8) are used: the visibility of each qubit is defined by

$$\mathscr{V}_k = \sum_{i \neq j} |\rho_{k_{ij}}|, \quad k = A, B \quad (6)$$

and the predictability reads

$$\mathscr{P}_k = |\rho_{k_{22}} - \rho_{k_{11}}|, \quad k = A, B. \quad (7)$$

Let $\Sigma = \sigma_y \otimes \sigma_y$ be the *spin flip* matrix and define $R(\rho) = \rho\Sigma\rho^*\Sigma$. The concurrence is given [10] by

$$\mathscr{C} = \max(0, \sqrt{r_1} - \sqrt{r_2} - \sqrt{r_3} - \sqrt{r_4}) \quad (8)$$

where $r_1 \geq r_2 \geq r_3 \geq r_4$ are the eigenvalues of $R(\rho)$.

## QND measurement circuits

In their paper, De Melo *et al.* [1] propose two quantum circuits for QND measurements on $|\chi\rangle$, which we will label circuits 1 and 2. Circuit 1, depicted in fig. 2, performs a nondemolition measurement of concurrence $\mathscr{C}$ on the two-qubit state $|\chi\rangle$ via the measurement of a third, ancilla qubit. As both circuits can measure concurrence, $\mathscr{C}$ is indexed to differentiate both cases: with circuit 1, it is computed as

$$\mathscr{C}_1 = |p_1 - p_0|, \tag{9}$$

where $p_0$ and $p_1$ are the the probabilities of outcome 0 and 1 respectively, while measuring the ancilla qubit in the computational basis.

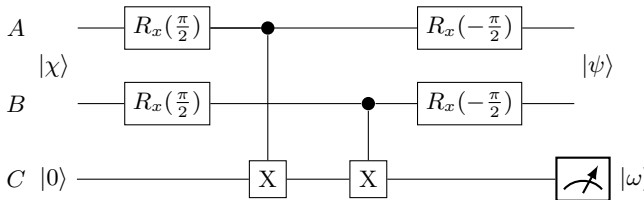

Fig. 2: Circuit 1 for the nondemolition measurement of the concurrence of the state $|\chi\rangle_{AB}$. Repeated measurements of the ancilla state $|\omega\rangle_C$ in the computational basis yield $p_0$ and $p_1$.

Circuit 2 is more versatile; it performs the nondemolition measurement of the two-qubit (non-local) concurrence of $|\chi\rangle$ as well as the one-qubit (local) coherence (i.e. visibility) $\mathscr{V}_k$ and predictability $\mathscr{P}_k$ of qubits $k = A, B$. It requires two ancillae qubits, labeled $C$ and $D$ (fig. 3).

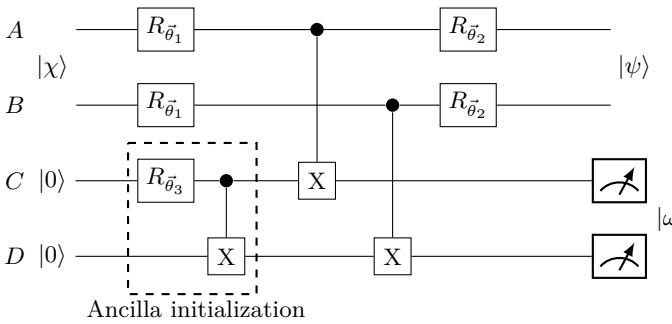

Ancilla initialization

Fig. 3: Circuit 2 for QND measurement of visibility, predictability and concurrence of the state $|\chi\rangle_{AB}$. Here, two ancillae qubits $C$ and $D$ are used and measured in the computational basis after interacting with $A$ and $B$ through C-NOT gates.

The choice of the quantity to be measured is done by applying the adequate single qubit rotations defined by

$$R_{\vec{\theta}_i} = e^{-i\vec{\sigma}\cdot\vec{\theta}_i}, \ i = 1, 2, 3. \tag{10}$$

where $R_{\vec{\theta}_{1,2}}$ act in parallel on $A$ and $B$ and $R_{\vec{\theta}_3}$ acts on $C$ before its entanglement with $D$. The measurement of the coherence of each qubit ($\mathscr{V}_A$, $\mathscr{V}_B$) is performed by choosing $\vec{\theta}_1 = -\vec{\theta}_2 = \frac{\pi}{2}\hat{y}$ and $\vec{\theta}_3 = 0$. One can measure the predictability ($\mathscr{P}_A$ and $\mathscr{P}_B$) by setting $\vec{\theta}_1 = \vec{\theta}_2 = \vec{\theta}_3 = 0$. Finally, for the concurrence $\mathscr{C}$, the circuit is specified by $\vec{\theta}_1 = -\vec{\theta}_2 = \frac{\pi}{2}\hat{x}$, $\vec{\theta}_3 = \frac{\pi}{2}\hat{y}$. After setting the angles $\{\theta_1, \theta_2, \theta_3\}$ corresponding to one of these three measurements (coherence, predictability or concurrence), the ancillae qubits evolve into the final state $|\omega\rangle_{CD}$, the measurements of which yield the probabilities $p_{ij}$ of qubits $C$ and $D$ outcoming in the $|i\rangle_C |j\rangle_D$ state $(i, j = 0, 1)$. From these results one can compute the quantities of interest for $A, B$ via

$$\mathscr{V}_A = |p_{00} + p_{01} - p_{10} - p_{11}|, \tag{11}$$

$$\mathscr{V}_B = |p_{00} + p_{10} - p_{01} - p_{11}| \tag{12}$$

when $\vec{\theta}_1 = -\vec{\theta}_2 = \frac{\pi}{2}\hat{y}$, $\vec{\theta}_3 = 0$, and

$$\mathscr{P}_A = |p_{00} + p_{01} - p_{10} - p_{11}|, \tag{13}$$

$$\mathscr{P}_B = |p_{00} + p_{10} - p_{01} - p_{11}| \tag{14}$$

when $\vec{\theta}_1 = \vec{\theta}_2 = \vec{\theta}_3 = 0$. The concurrence (conditional on $\vec{\theta}_1 = -\vec{\theta}_2 = \frac{\pi}{2}\hat{x}$, $\vec{\theta}_3 = \frac{\pi}{2}\hat{y}$) is given by

$$\mathscr{C}_2 = |p_{\Psi+} - p_{\Phi-}| \tag{15}$$

where $p_{\Psi+} = |\langle\Psi^+|\omega\rangle|^2$ and similarly for $p_{\Phi-}$, with $\Psi^+$ and $\Phi^-$ the Bell states defined above.

## Outgoing state

When measuring visibility, the 4-qubit state before the measurement of the ancillae qubits is found to be

$$\frac{1}{\sqrt{2}}\Big[(\eta - \beta)|-\rangle|-\rangle|00\rangle + (\alpha + \gamma)|-\rangle|+\rangle|01\rangle + \\ (\gamma - \alpha)|+\rangle|-\rangle|10\rangle + (\eta + \beta)|+\rangle|+\rangle|11\rangle\Big]. \tag{16}$$

Here we keep the conventions of [1], i.e. $|\pm\rangle = \frac{1}{\sqrt{2}}(|1\rangle \pm |0\rangle)$. When the circuit is set to measure predictability, the 4-qubit state reads

$$\frac{1}{\sqrt{2}}\Big[(\eta - \gamma)|00\rangle|00\rangle + (\beta - \alpha)|01\rangle|01\rangle + \\ (\alpha + \beta)|10\rangle|10\rangle + (\gamma + \eta)|11\rangle|11\rangle\Big] \tag{17}$$

and for concurrence,

$$\big(\alpha|\Psi^-\rangle + \eta|\Phi^+\rangle\big)|\Psi^+\rangle + \big(\gamma|\Phi^-\rangle + \beta|\Psi^+\rangle\big)|\Phi^+\rangle. \tag{18}$$

We stress that equation (16) is not equivalent to eq. (14) of ref. [1], which we slightly corrected (see appendix A).

Looking at eqs. (16) to (18) it is clear that when measuring the ancillae in the computational basis, the post-measurement state of the two-qubit system $A, B$ will maximize the value of the quantity which is measured on the input state; e.g. if the circuit measures the concurrence of the input state, the outgoing state after measuring the ancillae qubits will be maximally entangled ($\mathscr{C} = 1$) upon postselection of the ancillae outcomes (quantum state preparation criterion). Moreover, in the absence of decoherence, a consecutive measurement of the same observable is expected to yield identical results, satisfying the nondemolition criterion.

## EXPERIMENTAL RESULTS

We now present the experimental results obtained with the circuits introduced above, on different quantum computing architectures. For each observable, the two-qubit quantum state $|\chi\rangle$ is generated with a fixed value of $\theta$ for which the observable evolves as a periodic function of $\varphi$ which spans the range between its extremal values $(0, 1)$. One period of the evolution is investigated by uniformly stepping the values $\varphi_i$ with the increment $\varphi_{i+1} - \varphi_i = \frac{\pi}{32}$. The number of repetitions of the experiment (shots) for each set of parameters is set to 5000 for the measurement values to be converged.

We observed high fluctuations in the results of IBM Q, strongly depending on which of the 4 different IBM Q backends [21] (with different qubit coupling maps) was used. The experiments were also performed with auto-

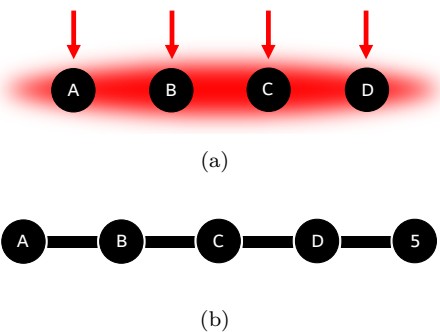

(a)

(b)

Fig. 4: Coupling between qubits of the quantum processors. (a) shows the ions which encode qubits in IonQ, arranged in line (details can be found in [22]). A global laser beam (horizontally spread spot) combined with pulses localized on each atom (arrows) enable two-qubit gates between any qubits of the chain. (b) shows the coupling map of ibmq_rome, which qubits are connected to their nearest neighbors via superconducting transmission lines.

matic "optimization" of the quantum circuit for the given backend [23]. Among our results on IBM Q, we found that the best overall performance was reached by the backend "ibmq_rome", with which, in addition, the optimization of the circuit generally enhanced the quality of the results. The corresponding data is therefore used to compare IBMQ with IonQ, in the following plots. The backend ibmq_rome holds 5 qubits and is one of the IBM Q machines in which each qubit is connected to its two nearest neighbors (fig. 4).

### Measurement accuracy and nondemolition criterion

The root of the mean squared (RMS) error of the measurements $\{\mathscr{O}_i^{\text{exp}}\}$ of an observable $\mathscr{O} = \mathscr{V}, \mathscr{P}, \mathscr{C}$ is defined as

$$E(\{\mathscr{O}_i^{\text{exp}}\}) = \sqrt{\frac{1}{N}\sum_{i=1}^{N}(\mathscr{O}(\varphi_i) - \mathscr{O}_i^{\text{exp}})^2}, \quad (19)$$

where $N$ is the total number of states spanned by the parameters $\varphi_i$, and $\{\mathscr{O}(\varphi_i)\}$ are the corresponding theoretically expected values of the observable.

First, the overall measurement accuracy of a particular implementation (QND measurement of one observable over one period) is estimated from the outcomes [21] of the QND measurements $\{\mathscr{O}_i^{\text{QND}}\}$ by computing the error $E(\{\mathscr{O}_i^{\text{QND}}\})$. This accuracy thus also depends on the quality of the input state preparation of $|\chi\rangle$. In parallel to the QND measurements, we performed tomographic measurements (see appendix B) directly on the input state $|\chi\rangle$ after the preparation circuit. These alternative measurements of the observable $\{\mathscr{O}_i^{\rho_\chi}\}$, performed for comparison, require less gates than the QND scheme and for this reason, are expected to be more accurate.

Second, the nondemolition criterion is verified in an equivalent way, via measurements of the same observable, $\{\mathscr{O}_i^{\rho_\psi}\}$, on the output state $|\psi\rangle$, via tomography, after the QND measurement.

The measurements of concurrence corresponding to these two steps are reported on fig. 5. In these plots, as well as in figs. 11 to 14 for other observables, results of measurements performed on the input state $|\chi\rangle$ with ibmq_rome are plotted with red circles, while the ones with IonQ are represented by blue squares. QND measurements are reported with filled symbols, while empty ones are used for tomographic measurements. Measurements on the output state are plotted with cross marks ($\times$ for ibmq_rome and $+$ for IonQ). As the second circuit is deeper and requires more gates than the first one, the observed drop of the QND-measured concurrence is unsurprising, even though it is more pronounced with ibmq_rome than with IonQ. Entanglement being sensitive to interactions with the environment [24], we observe a large deviation between QND measurement and theoretical value for highly entangled input states. A simple scaling of the theoretical curve can be used to fit the data; however this fitting method fails when applied to

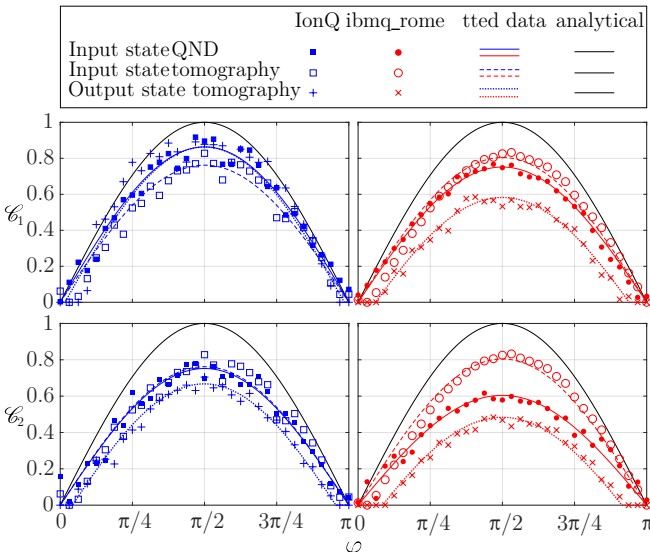

Fig. 5: The concurrence of input state $|\chi\rangle$ (with $\theta = \pi$, $\lambda = 0$) measured by the QND circuits 1 (top row) and 2 (bottom row) is shown with full symbols. Results from ibmq_rome are shown on the right column, those from IonQ on the left. The concurrence for qubits $A, B$ computed from their state tomography at the input and the output of the circuit is shown as open and crossed symbols, respectively. Color code is red for ibmq_rome and blue for IonQ. The measurements are fitted with the theoretical curve multiplied by a scaling factor. For the outgoing state $|\psi\rangle$, the fits are theoretical states to which a fully mixed component was added. Note that the tomographic measurement of $|\chi\rangle$ is independent of the QND circuit, and the same data is reported on the top and bottom graphs (empty squares and circles).

the output state tomography $|\psi\rangle$, because concurrence is affected by the drop in state purity after the QND circuit [25]. We therefore obtain a good fit to the data by introducing a fully mixed component in the theoretical state (dotted lines in fig. 5). Measurements for the single-partite observables $\mathscr{V}_A$, $\mathscr{V}_B$, $\mathscr{P}_A$, $\mathscr{P}_B$ are discussed in appendix C and illustrated in figs. 11 to 14 .

The RMS error of all the QND measurements shown in figs. 5 and 11 to 14, and their repetition on additional backends, are gathered in table II. IonQ did perform better than all IBM Q backends, except for the QND measurement of $\mathscr{V}_B$ and $\mathscr{P}_B$, for which ibmq_vigo was the best. Comparing the performance of the QND measurement and the tomographic measurement (table I in appendix B), one can see that for ibmq_vigo, the QND measurement does outperform the tomographic measurement, which is not expected (the tomographic measurement does not require any two-qubit gate) and thus probably reflects the fluctuation in the quality of the preparation of $|\chi\rangle$. This is the reason why we prefer the analytical computation to the parallel tomographic measurement when it comes to evaluating the accuracy of the

QND measurement, in addition to the readout errors.

The nondemolition of the observable is characterized by the error of the tomographic measurement on the output state with respect to the theoretical expected value (table III, computed with the same procedure than table II). Here again, IonQ outperforms most IBM Q backends.

Finally, the mean value of all the measurement errors obtained on the two systems (eq. (19) averaged over all experiments and all observables) is shown with color bars in fig. 6. We observe that the performance of IBM Q

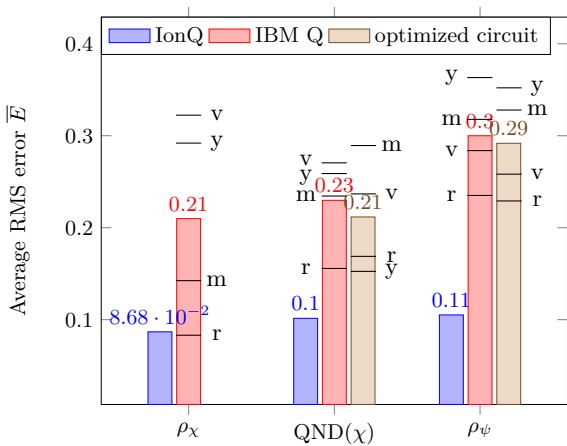

Fig. 6: Averaged RMS error $\overline{E}$ over all experiments on IonQ and IBM Q computers, for the three measurement steps. Input state tomography $\rho_\chi$: $\overline{E(\{\mathscr{O}_i^{\rho_\chi}\})}$, QND measurement of $|\chi\rangle$: $\overline{E(\{\mathscr{O}_i^{\text{QND}}\})}$, and output state tomography $\rho_\psi$: $\overline{E(\{\mathscr{O}_i^{\rho_\psi}\})}$. IBM Q backends are labeled r: ibmq_rome, v: ibmq_vigo, y: ibmq_5_yorktown, m: ibmq_16_melbourne.

changes consequently from a backend to another (dispersion of the error in fig. 6). After averaging, as expected, the error of the tomographic measurement is smaller than the one of the QND measurement, itself smaller than error of the post-QND tomographic measurement: $\overline{E(\{\mathscr{O}_i^{\rho_\chi}\})} < \overline{E(\{\mathscr{O}_i^{\text{QND}}\})} < \overline{E(\{\mathscr{O}_i^{\rho_\psi}\})}$. The advantage of IonQ on IBM Q is particularly consequent for the nondemolition criterion. Indeed, with IBM Q the error is 34% higher on the output state measurement than the QND, whereas with IonQ this ratio equals 4%. Our averaged data also shows a relatively small reduction of the error by the optional optimization of the circuit.

## Quantum state preparation

In order to check the quantum state preparation criterion, we perform an additional tomography of the state $|\psi\rangle$ after the QND measurement, allowing to measure the same observable as the QND circuit, on the conditional output state (i.e. while postselecting results among the outcomes of $|\omega\rangle$, measuring the ancillae qubits in the

computational basis). Calculations show that after the measurement of a given quantity by the circuit, this very same quantity will equal unity for the outgoing state. We stress that there are values of the parameters $(\varphi, \theta)$ of the preparation circuit (fig. 1) for which certain outcomes of $|\omega\rangle$ have low or vanishing probability. For example, measuring the visibility, the setting $(\phi = \pi/2, \theta = 0)$ implies $(\eta - \beta) = (\alpha + \gamma) = 0$, thus no possibility to postselect the ancilla states $|\omega\rangle = |00\rangle$ and $|\omega\rangle = |01\rangle$. The probability amplitudes of the quantum states of eqs. (16) to (18) are represented as a function of $\varphi$ in fig. 16 (appendix C). Postselecting states with a low number of outcomes causes poor density matrix estimation (low fidelity) and the observables cannot be retrieved accurately. Our experimental procedure generates 5000 states per circuit, and provides a sufficient number of outcomes to reach a converged fidelity of postselected states, which stays constant for a given range of $\varphi$, as visible in fig. 7. In particular, we experimentally observe that a probability amplitude above 0.5 (see fig. 16) is sufficient to measure the conditional density matrix without degrading the fidelity $F$, a measure of distance between the experimental density matrix $\rho_{\mathrm{exp}}$ and the theoretical expected one $\rho_{\mathrm{th}}$, defined as

$$F = \mathrm{Tr}\left( \sqrt{\sqrt{\rho_{\mathrm{th}}}\rho_{\mathrm{exp}}\sqrt{\rho_{\mathrm{th}}}} \right)^2. \qquad (20)$$

The quantum state preparation of maximally entangled states $|(\psi|_{\omega=1})\rangle = |\Phi^+\rangle$ with the circuit of fig. 2 is reported on fig. 7, showing the concurrence and the fidelity of the postselected output states. Figure 8 shows the

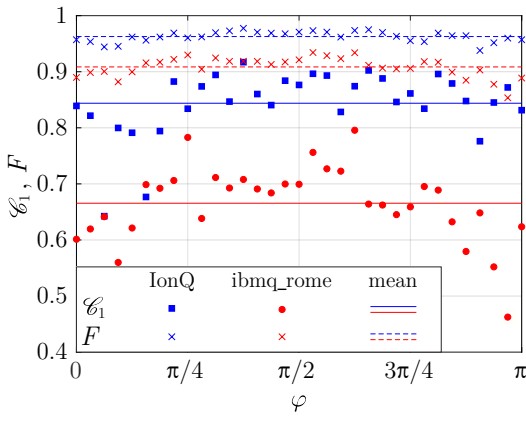

Fig. 7: Concurrence and fidelity measured on the conditional output state $|(\psi|_{\omega=1})\rangle$ using the circuit of fig. 2. The measured concurrence is $0.84 \pm 0.08$ on IonQ and $0.67 \pm 0.21$ on ibmq_rome. On IonQ $F = 0.96 \pm 0.02$ while ibmq_rome reaches $F = 0.91 \pm 0.05$.

state preparation related to each observable for the circuit of fig. 3, for ibmq_rome [26] and IonQ. Among these results, some states were postselected for outcomes of $|\omega\rangle$ with a probability amplitude under 0.5: $\mathscr{V}_A$ (near $\varphi = 0$

and $\varphi = \pi$) and $\mathscr{V}_B$ (near $\varphi = 0$), see fig. 8. The consequent degradation of the measurement is particularly apparent on ibmq_rome.

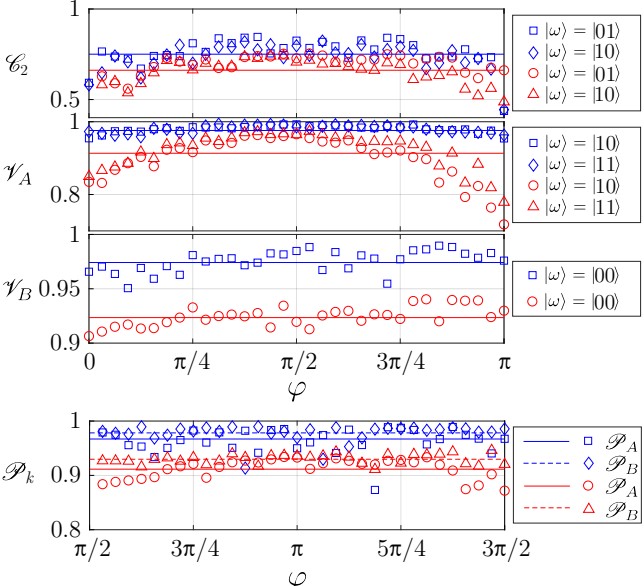

Fig. 8: Conditional measurements of $\mathscr{C}_2$, $\mathscr{V}_A$, $\mathscr{V}_B$, and $\mathscr{P}_{A,B}$ ($|\omega\rangle = |00\rangle$) on the output state for the circuit of fig. 3. Mean values (horizontal lines) are reported in table V.

The mean value of each observable for the states displayed in figs. 7 and 8 are reported in table V, confirming the better performance of IonQ with respect to IBM Q for state preparation. Our measurements show that the quality of the state preparation by IonQ was superior for each experiment (highlighted in table V). Averaging over each observable and each IBM Q backend, the values of the measured observables on conditional states $|(\psi|_\omega)\rangle$ are 0.914 for IonQ and 0.679 (0.691) for IBM Q (with circuit optimization).

### Fidelity measurements

We computed the state fidelity (20) of $|\chi\rangle$, $|\psi\rangle$ and $|(\psi|_\omega)\rangle$ (see tables VI to VIII in appendix E). In general, the state fidelity drops with postselection on IBM Q, and stays constant on IonQ: fig. 9. Here again, we observe changing fidelity with the IBM Q backends, and also that the circuit optimization does improve the result for some backends, whereas for others it leaves it unchanged.

We noticed that, for maximally entangled states, for which high fidelity of preparation is challenging, the fidelity is actually increased by postselection. We repeated the experiment 50 times for the input state $|\chi\rangle = |\Phi^+\rangle$, and measured concurrence and fidelity in the output, with and without postselection. The result, for the circuits of figs. 2 and 3, is shown in figs. 10 and 18 (see

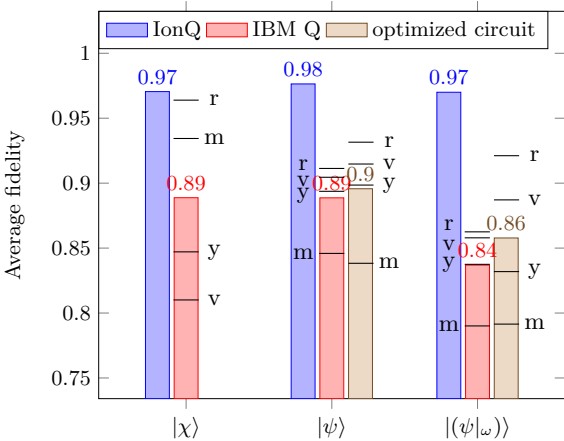

Fig. 9: Averaged fidelity of all states measured via tomography on IonQ and IBM Q, for the input states $|\chi\rangle$, output states $|\psi\rangle$ and postselected output states $|(\psi|_\omega)\rangle$.

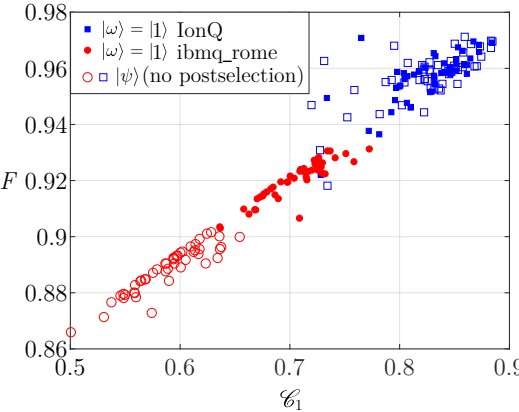

Fig. 10: Fidelity as a function of concurrence measured on the output state, using the circuit of fig. 2. 50 repetitions using the input state $|\chi\rangle = |\Phi^+\rangle$.

appendix E). Note that here again the difference is particularly visible for IBM Q, while quite constant on IonQ.

## CONCLUSION

We reported a first implementation of the scheme proposed by De Melo *et al.* [1] designed for the QND measurement of complementary observables on a bipartite quantum system, using online available quantum computers. We employed processors based on two different technologies, namely superconducting qubits on four machines proposed by IBM Q, and trapped-ion qubits on IonQ. In the context of this experiment, we found that trapped-ion qubits produced measurement outcomes closest to those expected from an ideal circuit, which corroborates past studies [22, 27]. Outcomes from IBM Q circuits showed in general less reproducibility over time. We observed important variations in the the results ob-

tained on IBM Q depending on the backend, the choice of which also turned out to greatly influence the efficiency of the circuit optimization routine. However, numerous other aspects reflecting the quality of a quantum computer for different tasks were not addressed in our study.

Looking ahead, scalability remains a challenge for both platforms. The computation time is in general much longer for trapped-ion based quantum bits, which however feature a much longer coherence time with respect to superconducting ones – but progress on the speed of entangling operations on ion qubits was recently reported, see e.g. [28]. Another key difference between these two technologies is the connectivity between qubits. While superconducting chips require a physical transmission line to connect two qubits, trapped ions can realize a fully connected set of qubits, which drastically eases the mapping of a quantum circuit to physical qubits, with specific methods to this end (see e.g. [29]). For instance, while experimenting on ibmq_rome, using circuit 2 and measuring concurrence, the actual transpiled circuit contained 6 C-NOT gates. This mapping is a central task when using NISQ computers, because of the lack of efficient error correction. If the full connectivity of the IonQ computer probably enhances the fidelity of our measurements, it would have a greater impact when generalizing this experiment to a larger number of qubits, as in recent works [30, 31]. Finally, we note that future studies will be eased by the recently announced possibility to access IonQ via Qiskit [32].

## ACKNOWLEDGEMENTS

We thank Dr. Marc-André Dupertuis for useful advice, comments and discussions. We acknowledge use of IBM Quantum, and Amazon Web Services' quantum computing service Amazon Braket. We thank very much the Amazon Braket team for their technical help. The views expressed are those of the authors and do not reflect the official policy or position of IBM, the IBM Q team, Amazon or Amazon Web Services.

## DATA AVAILABILITY STATEMENT

The code that supports this study is openly available in `GitHub` at `https://github.com/NicoSchwaller/QND-measurement-of-complementarity`.

* nicolas.schwaller@epfl.ch
† valeria.vento@epfl.ch
‡ chris.galland@epfl.ch

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

## Appendix A: Check for (16)

Let $\rho_i$ be the 4-qubit state after state preparation by the circuit of fig. 1,

$$\rho_i = |\chi\rangle\langle\chi| \otimes |0\rangle\langle 0|, \tag{21}$$

the two-qubit identity operator $\mathbb{1} = \begin{pmatrix} 1 & 0 \\ 0 & 1 \end{pmatrix}$, and the rotation along $\hat{y}$,

$$R = \begin{pmatrix} \cos\frac{\pi}{4} & -\sin\frac{\pi}{4} \\ \sin\frac{\pi}{4} & \cos\frac{\pi}{4} \end{pmatrix}. \tag{22}$$

The three-qubit C-NOT gate with first qubit as control and last one as target is written in the usual computa-

tional basis as

$$\text{CNOT}_{1\to3} = \begin{pmatrix} 1 & 0 & 0 & 0 & 0 & 0 & 0 & 0 \\ 0 & 1 & 0 & 0 & 0 & 0 & 0 & 0 \\ 0 & 0 & 1 & 0 & 0 & 0 & 0 & 0 \\ 0 & 0 & 0 & 1 & 0 & 0 & 0 & 0 \\ 0 & 0 & 0 & 0 & 0 & 1 & 0 & 0 \\ 0 & 0 & 0 & 0 & 1 & 0 & 0 & 0 \\ 0 & 0 & 0 & 0 & 0 & 0 & 0 & 1 \\ 0 & 0 & 0 & 0 & 0 & 0 & 1 & 0 \end{pmatrix}. \qquad (23)$$

The evolution of $\rho_i$ through the QND circuit of fig. 3 is described by the operator

$$U_V = (R^{-1} \otimes R^{-1} \otimes \mathbb{1} \otimes \mathbb{1})(\mathbb{1} \otimes \text{CNOT}_{1\to3})$$
$$(\text{CNOT}_{1\to3} \otimes \mathbb{1})(R \otimes R \otimes \mathbb{1} \otimes \mathbb{1}).$$

It is easy to check that $U_V \rho_i U_V^\dagger$ is the state (16).

## Appendix B: Quantum state tomography

Quantum state tomography is a procedure by which one can infer the density matrix, and requires a complete set of measurements (as many as the number of real parameters in the density matrix). The density matrix holds all the information needed to compute the expectation values of any system observable. In particular, concurrence, coherence and predictability in a two qubit system can be computed from its density matrix, directly using eqs. (6) to (8).

On IBM Q machines, a built-in method for quantum state tomography is used: the function "state_tomography_circuits" sets up the circuits needed for two-qubit state tomography, and "StateTomography-Fitter" does compute the density matrix using maximum likelyhood methods. This technique allows to increase the speed of tomographic measurements, by reducing the number of required circuits. On the IonQ quantum processor, we implement the linear tomography method presented in [33] which consists in the execution of 16 circuits per tomography step. Those two methods, albeit different, are both valid to measure the density matrix of the system. Table I reports the errors of the tomographic measurements (with respect to the theoretically expected value) performed on the state $|\chi\rangle$. These tomographic measurements of the input states are used as reference measurements to which one can compare the QND method. The values reported in table I show that the quality of the state preparation is not constant: the tomographic measurement is expected to be slightly more efficient than the QND measurement (see appendix C, table II), in particular because tomography requires less entangling gates. However, significantly higher errors are sometimes observed for the initial tomographic measurement. Up to some point, this could also be due to noise in the measurement process, but in some cases most probably to poor state preparation.

| | $\mathscr{V}_A$ | $\mathscr{V}_B$ | $\mathscr{P}_A$ | $\mathscr{P}_B$ | $\mathscr{C}$ |
|---|---|---|---|---|---|
| **IonQ** | 0.017 | 0.126 | 0.054 | 0.062 | 0.175 |
| **ibmq_rome** | 0.04 | 0.076 | 0.072 | 0.082 | 0.144 |
| **ibmq_vigo** | 0.043 | 0.263 | 0.507 | 0.123 | 0.676 |
| **ibmq_5_yorktown** | 0.147 | 0.192 | 0.391 | 0.234 | 0.496 |
| **ibmq_16_melbourne** | 0.043 | 0.105 | 0.182 | 0.134 | 0.247 |

Tab. I: Root of mean squared error of the tomographic measurements on the input state, with respect to the expected values, $E(\{\mathscr{O}_i^{\rho_\chi}\})$. Best performance highlighted for each quantity.

## Appendix C: QND and output state measurements

The figures of this section report the QND and tomographic measurements of visibility and predictability. The data is plotted as in fig. 5: blue squares for IonQ, red circles for IBM Q, which are filled (QND) and empty (tomography). Crosses are used for the output state.

In opposition to concurrence, as piloting $\mathscr{V}_A$ requires only the local rotation defined by the angle $\varphi$, the state preparation is more efficient than for entangled states (fig. 11). In fact, one can see that the QND measure-

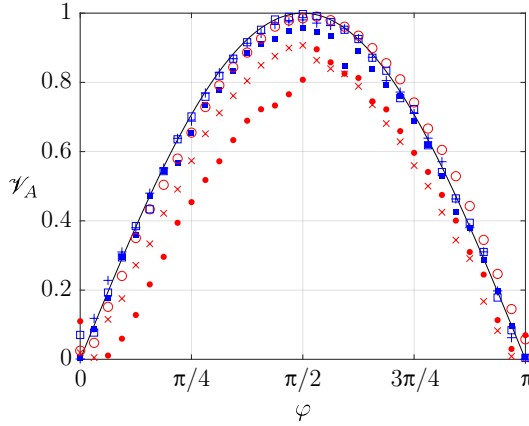

Fig. 11: Visibility $\mathscr{V}_A$ of input state $|\chi\rangle$ ($\theta = 0$) measured by the QND circuit (filled symbols) and the state tomography (empty symbols), using ibmq_rome (red) and IonQ (blue) quantum bits, and measurements of $\mathscr{V}_A$ on the output state $|\psi\rangle$ (crosses). The symbols follow the convention of fig. 5.

ment gives an excellent measurement of $\mathscr{V}_A$ on IonQ. The measurement on the output state is also very close to the expected value, for the full range of analyzed states. While the state preparation on IBM Q is satisfying (input state tomography, empty red circles on fig. 11), a significant gap to the expected result appears for the QND measurement and the output state characterization on ibmq_rome. Varying the visibility of the qubit $B$ with the state preparation circuit of fig. 1 requires a nonzero value of $\theta$, a more costly operation than when studying $\mathscr{V}_A$. The measurements of $\mathscr{V}_B$ (fig. 12) seem more noisy for IonQ than IBM Q, and it cannot be determined visually which machine gives the most accurate result. How-

ever the peak value of $\mathscr{V}_B$ at $\varphi = \frac{\pi}{32}$, IonQ seems to perform better.

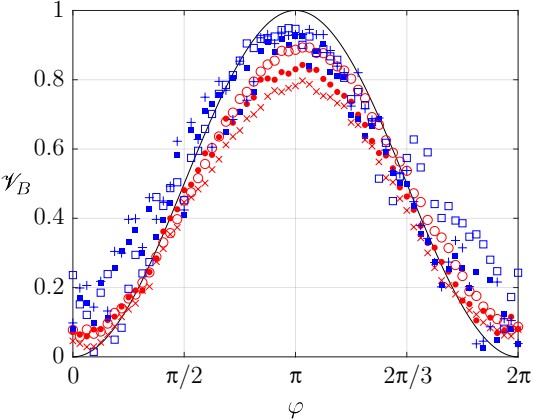

Fig. 12: Visibility $\mathscr{V}_B$ of input state $|\chi\rangle$ ($\theta = \frac{3\pi}{2}$) measured by the QND circuit (fig. 3) and the state tomography, using ibmq_rome and IonQ quantum bits, and measurements of $\mathscr{V}_B$ on the output state $|\psi\rangle$ (crosses). The symbols follow the convention of fig. 5.

Coming to predictability, we measure $\mathscr{P}_A$ and $\mathscr{P}_B$ on the same quantum states (i.e. the same values of $\varphi, \theta$). This enables to compare values ($\mathscr{P}_A$ and $\mathscr{P}_B$) truly measured at the same time on the same quantum system.

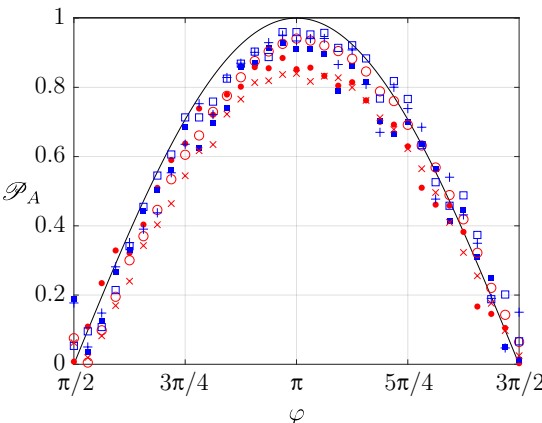

Fig. 13: Predictability $\mathscr{P}_A$ of input state $|\chi\rangle$ ($\theta = \pi$) measured by the QND circuit (filled symbols) and the state tomography, using ibmq_rome (red) and IonQ (blue) quantum bits, and measurements of $\mathscr{P}_A$ on the output state $|\psi\rangle$ (crosses). The symbols follow the convention of fig. 5.

On IBM Q (red data in figs. 13 and 14), an asymmetric behavior is observed: the measured predictability of the qubit $B$ is more distant from the expected value than the one of qubit $A$. One can clearly see a higher deviation in the QND and final tomography measurements of $\mathscr{P}_B$ on IBM Q for qubit $B$. This is the case around $\varphi = \pi$, where $|\chi\rangle = |11\rangle$. The excited state of the second qubit ($B$) is prepared through the C-NOT gate, a process which may be responsible for the degradation of the state of qubit $B$ with respect to qubit $A$.

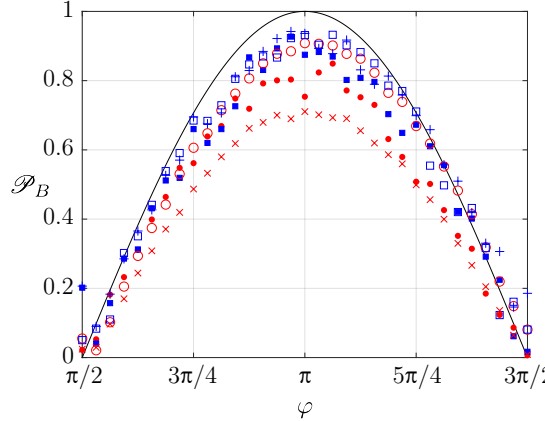

Fig. 14: Predictability $\mathscr{P}_B$ of input state $|\chi\rangle$ ($\theta = \pi$) measured by the QND circuit (filled symbols) and the state tomography (empty symbols), using ibmq_rome (red) and IonQ (blue) quantum bits, and measurements of $\mathscr{P}_B$ on the output state $|\psi\rangle$ (crosses). The symbols follow the convention of fig. 5.

The mean error between the measurements and the theoretically expected values are reported for each machine and for each observable in table II (QND measurements) and table III (output state tomographic measurements).

| | $\mathscr{V}_A$ | $\mathscr{V}_B$ | $\mathscr{P}_A$ | $\mathscr{P}_B$ | $\mathscr{C}_1$ | $\mathscr{C}_2$ |
|---|---|---|---|---|---|---|
| **IonQ** | 0.039 | 0.102 | 0.085 | 0.096 | 0.107 | 0.18 |
| **ibmq_rome** | 0.169 | 0.12 | 0.11 | 0.141 | 0.175 | 0.218 |
| **with optimization** | 0.177 | 0.096 | 0.092 | 0.139 | 0.23 | 0.278 |
| **ibmq_vigo** | 0.171 | 0.141 | 0.124 | 0.455 | 0.215 | 0.517 |
| **with optimization** | 0.261 | 0.063 | 0.33 | 0.068 | 0.129 | 0.569 |
| **ibmq_5_yorktown** | 0.107 | 0.340 | 0.189 | 0.242 | 0.119 | 0.556 |
| **with optimization** | 0.139 | 0.088 | 0.155 | 0.105 | 0.148 | 0.279 |
| **ibmq_16_melbourne** | 0.15 | 0.117 | 0.242 | 0.2 | 0.274 | 0.423 |
| **with optimization** | 0.147 | 0.336 | 0.151 | 0.237 | 0.293 | 0.572 |

Tab. II: Root of mean squared error of the QND measurements on the input state $E(\{\mathscr{O}_i^{\mathrm{QND}}\})$. The best measurement accuracy is highlighted for each quantity.

| | $\mathscr{V}_A$ | $\mathscr{V}_B$ | $\mathscr{P}_A$ | $\mathscr{P}_B$ | $\mathscr{C}_1$ | $\mathscr{C}_2$ |
|---|---|---|---|---|---|---|
| **IonQ** | 0.015 | 0.106 | 0.076 | 0.078 | 0.096 | 0.26 |
| **ibmq_rome** | 0.059 | 0.229 | 0.223 | 0.237 | 0.327 | 0.335 |
| **with optimization** | 0.119 | 0.126 | 0.125 | 0.208 | 0.398 | 0.398 |
| **ibmq_vigo** | 0.225 | 0.098 | 0.229 | 0.147 | 0.317 | 0.687 |
| **with optimization** | 0.161 | 0.093 | 0.231 | 0.149 | 0.285 | 0.63 |
| **ibmq_5_yorktown** | 0.399 | 0.222 | 0.224 | 0.202 | 0.533 | 0.6 |
| **with optimization** | 0.346 | 0.219 | 0.254 | 0.168 | 0.531 | 0.595 |
| **ibmq_16_melbourne** | 0.029 | 0.138 | 0.211 | 0.182 | 0.651 | 0.696 |
| **with optimization** | 0.028 | 0.171 | 0.208 | 0.178 | 0.687 | 0.696 |

Tab. III: Root of mean squared error of the tomographic measurement on the output state $E(\{\mathscr{O}_i^{\rho_\psi}\})$. The most efficient nondemolition of the observable is highlighted for each quantity.

In table IV and fig. 15 we show the error between the value of the observable measured via QND, and the one obtained by the following destructive tomography. We expect the decoherence happening between the two mea-

surements to increase the error from the QND to the tomography, i.e. positive values in table IV and fig. 15.

| | $\mathscr{V}_A$ | $\mathscr{V}_B$ | $\mathscr{P}_A$ | $\mathscr{P}_B$ | $\mathscr{C}_1$ | $\mathscr{C}_2$ |
|---|---|---|---|---|---|---|
| **IonQ** | -0.024 | 0.004 | -0.009 | -0.018 | -0.011 | 0.08 |
| **ibmq_rome** | -0.11 | 0.109 | 0.113 | 0.096 | 0.152 | 0.117 |
| **with optimization** | -0.058 | 0.03 | 0.033 | 0.069 | 0.162 | 0.12 |
| **ibmq_vigo** | 0.054 | -0.043 | 0.105 | -0.308 | 0.102 | 0.17 |
| **with optimization** | -0.1 | 0.03 | -0.099 | 0.081 | 0.156 | 0.061 |
| **ibmq_5_yorktown** | 0.292 | -0.118 | 0.035 | -0.04 | 0.414 | 0.044 |
| **with optimization** | 0.207 | 0.131 | 0.099 | 0.063 | 0.383 | 0.316 |
| **ibmq_16_melbourne** | -0.121 | 0.021 | -0.031 | -0.018 | 0.377 | 0.273 |
| **with optimization** | -0.119 | -0.165 | 0.057 | -0.059 | 0.394 | 0.124 |

Tab. IV: Mean error between the QND measurement and the subsequent tomographic estimation, $E(\{\mathscr{O}_i^{\rho_\psi}\}) - E(\{\mathscr{O}_i^{\text{QND}}\})$. A negative value indicates that result of the second one was closer to the theoretically expected value.

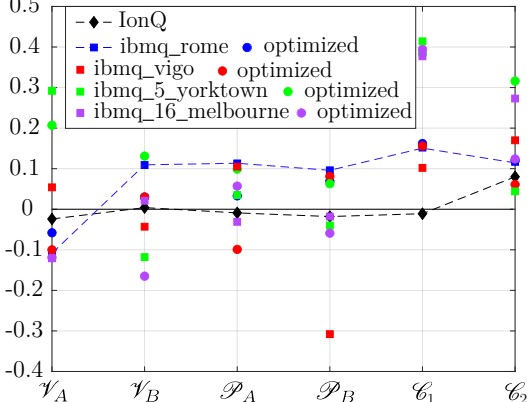

Fig. 15: Mean error between the QND measurement and the subsequent tomographic estimation, $E(\{\mathscr{O}_i^{\rho_\psi}\}) - E(\{\mathscr{O}_i^{\text{QND}}\})$. Dashed lines are traced between results of IonQ, which most of the time provide the smallest error, and ibmq_rome, the best performing IBM Q backend.

Negative values appear when the measurement noise is high compared to the degradation of the state between the two measurement steps. Note that the case of $\mathscr{C}_2$ (followed by $\mathscr{C}_1$) is the measurement for which the state is most likely to experience decoherence, and in that case we observe an actual degradation from the QND to the tomographic measurement for every machine: fig. 15. One can see that using IonQ, the two measurements are relatively close (black dashed line on fig. 15). Indeed, as showed in fig. 6, the quantum state is robust to decoherence between the two measurements on IonQ. On IBM Q, we observed considerably higher degradation of the state between the two steps, but also some unexpectedly high measurement errors. The backend ibmq_rome seems not to be subject to these errors, thus we can observe the effective degradation of the state between the measurements (blue dashed line in fig. 15). Finally, we may remark that the optimization of the quantum circuits on IBM Q does not result in any notable improvement in most cases.

## Appendix D: quantum state preparation criterion

Figure 16 shows the probability amplitudes of the quantum states of eqs. (16) to (18). The states with sufficiently high probability amplitude (see figs. 7 and 8) are postselected before measurement of the observables, i.e. using the QND measurement as a state preparation circuit. The average value measured on those states is reported in table V for each observable.

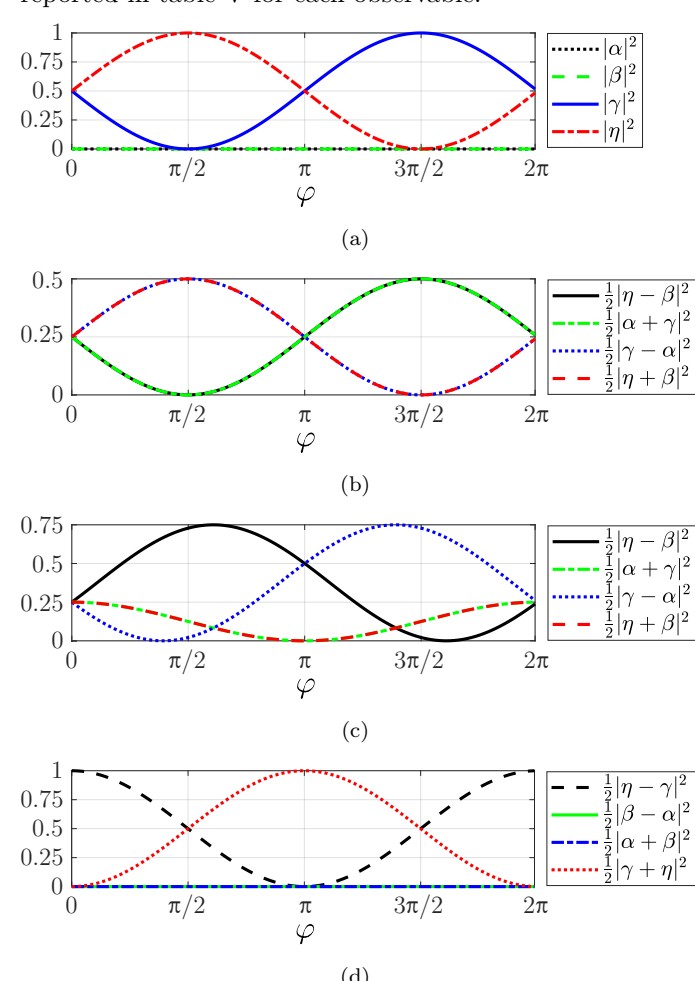

Fig. 16: Probability amplitudes of the output states of eqs. (16) to (18) for the measurement settings of (a) $\mathscr{C}$ with $\theta = \pi$ (b) $\mathscr{V}_k$ with $\theta = 0$ (c) $\mathscr{V}_k$ with $\theta = 3\pi/2$ and (d) $\mathscr{P}_k$ with $\theta = \pi$.

| | $\mathscr{V}_A$ | $\mathscr{V}_B$ | $\mathscr{P}_A$ | $\mathscr{P}_B$ | $\mathscr{C}_1$ | $\mathscr{C}_2$ |
|---|---|---|---|---|---|---|
| **IonQ** | 0.98 | 0.975 | 0.964 | 0.977 | 0.844 | 0.746 |
| **ibmq_rome** | 0.899 | 0.903 | 0.726 | 0.728 | 0.666 | 0.656 |
| **with optimization** | 0.812 | 0.925 | 0.91 | 0.93 | 0.54 | 0.577 |
| **ibmq_vigo** | 0.926 | 0.753 | 0.886 | 0.83 | 0.674 | 0.047 |
| **with optimization** | 0.933 | 0.783 | 0.925 | 0.885 | 0.709 | 0.17 |
| **ibmq_5_yorktown** | 0.388 | 0.668 | 0.837 | 0.84 | 0.419 | 0.258 |
| **with optimization** | 0.494 | 0.626 | 0.734 | 0.833 | 0.422 | 0.246 |
| **ibmq_16_melbourne** | 0.96 | 0.828 | 0.866 | 0.869 | 0.181 | 0.475 |
| **with optimization** | 0.966 | 0.786 | 0.849 | 0.87 | 0.142 | 0.504 |

Tab. V: Mean value of the measured observable on the conditional states $|\psi\rangle$, postselected as in figs. 7 and 8. The best state preparation is highlighted for each quantity.

## Appendix E: Fidelity measurements

We computed the mean fidelity (20) of all relevant states $|\chi\rangle$ ($|\psi\rangle$) for each observable, and reported it in table VI (table VII). The fidelity of the postselected

|  | $\mathscr{V}_A$ | $\mathscr{V}_B$ | $\mathscr{P}_{A,B}$ | $\mathscr{C}$ |
|---|---|---|---|---|
| **IonQ** | 0.996 | 0.958 | 0.966 | 0.962 |
| **ibmq_rome** | 0.984 | 0.956 | 0.956 | 0.959 |
| **ibmq_vigo** | 0.98 | 0.754 | 0.765 | 0.741 |
| **ibmq_5_yorktown** | 0.914 | 0.845 | 0.803 | 0.826 |
| **ibmq_16_melbourne** | 0.979 | 0.927 | 0.905 | 0.926 |

Tab. VI: Mean fidelity of the states $|\chi\rangle$ generated for the measurement of each observable.

|  | $\mathscr{V}_A$ | $\mathscr{V}_B$ | $\mathscr{P}_{A,B}$ | $\mathscr{C}_1$ | $\mathscr{C}_2$ |
|---|---|---|---|---|---|
| **IonQ** | 0.998 | 0.99 | 0.972 | 0.987 | 0.935 |
| **ibmq_rome** | 0.998 | 0.969 | 0.77 | 0.914 | 0.905 |
| **with optimization** | 0.993 | 0.98 | 0.907 | 0.886 | 0.891 |
| **ibmq_vigo** | 0.979 | 0.98 | 0.91 | 0.921 | 0.732 |
| **with optimization** | 0.987 | 0.976 | 0.917 | 0.912 | 0.781 |
| **ibmq_5_yorktown** | 0.899 | 0.953 | 0.902 | 0.857 | 0.857 |
| **with optimization** | 0.93 | 0.955 | 0.895 | 0.857 | 0.855 |
| **ibmq_16_melbourne** | 0.999 | 0.977 | 0.905 | 0.721 | 0.627 |
| **with optimization** | 0.998 | 0.975 | 0.906 | 0.689 | 0.623 |

Tab. VII: Mean fidelity of the states $|\psi\rangle$ after the QND measurement.

states is given in table VIII. Fidelity tends to decrease with the postselection (see fig. 9 which summarizes tables VI to VIII). To illustrate this fact, fig. 17 reports

|  | $\mathscr{V}_A$ | $\mathscr{V}_B$ | $\mathscr{P}_{A,B}$ | $\mathscr{C}_1$ | $\mathscr{C}_2$ |
|---|---|---|---|---|---|
| **IonQ** | 0.978 | 0.983 | 0.986 | 0.963 | 0.94 |
| **ibmq_rome** | 0.958 | 0.917 | 0.635 | 0.909 | 0.893 |
| **with optimization** | 0.946 | 0.946 | 0.96 | 0.868 | 0.885 |
| **ibmq_vigo** | 0.858 | 0.91 | 0.928 | 0.913 | 0.68 |
| **with optimization** | 0.904 | 0.916 | 0.954 | 0.914 | 0.747 |
| **ibmq_5_yorktown** | 0.771 | 0.86 | 0.921 | 0.828 | 0.807 |
| **with optimization** | 0.778 | 0.849 | 0.897 | 0.83 | 0.805 |
| **ibmq_16_melbourne** | 0.933 | 0.914 | 0.934 | 0.61 | 0.559 |
| **with optimization** | 0.937 | 0.904 | 0.93 | 0.64 | 0.546 |

Tab. VIII: Mean fidelity of the conditional states $|\psi\rangle$ postselected as in figs. 7 and 8 after the measurement of each observable.

the repetition of the experiment for the measurement of maximal $\mathscr{V}_B$ state ($\theta = \frac{3\pi}{2}, \varphi = \pi$). As observed in general, the mean fidelity is slightly decreased by postselection on ibmq_rome, and close to unchanged on IonQ. In the end, it is clear that postselection does perform efficient state preparation, even healing the measured value of an observable in the case of an incoming eigenstate. Indeed, in the case of the measurement of concurrence

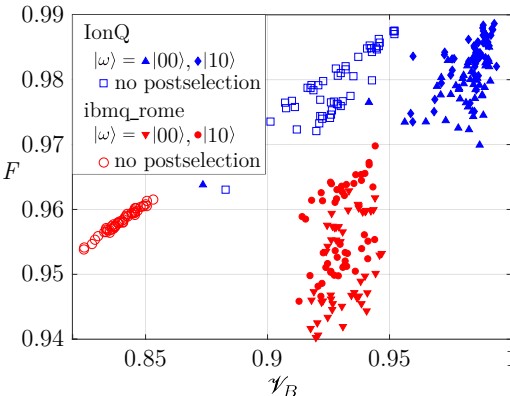

Fig. 17: Fidelity as a function of $\mathscr{V}_B$ measured on the output state, using the circuit of fig. 3.

with the circuit of fig. 5 ($\mathscr{C}_1$), IonQ provides sufficient state fidelity that postselection does not results in higher concurrence, in opposition to ibmq_rome, on which postselection clearly purifies entanglement (fig. 10). In the case of $\mathscr{C}_2$, for which the state is even more fragile and affected by decoherence, postselection happens to be useful on both systems (fig. 18), clearly increasing the state fidelity together with the target value of the observable.

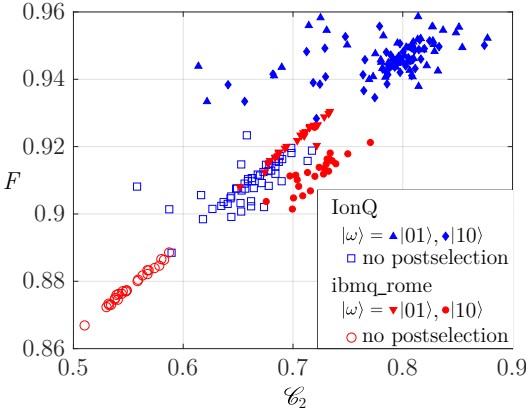

Fig. 18: Fidelity as a function of concurrence measured on the output state, using the circuit of fig. 3. 50 repetitions using the input state $|\chi\rangle = |\Phi^+\rangle$.

We observed that the difference in the tomography procedure used on IBM Q and IonQ (see appendix B) is the main contribution to the higher noise (thicker spreading of the measurements along the line) for IonQ in the results of figs. 10, 17 and 18. In fact, in opposition with IBM Q, the method we used on IonQ produces matrices that are not necessarily positive semi-definite. They are nevertheless valid estimations of the density matrix of the considered two-qubit state, and the aforementioned artifact does not hinder comparisons of the mean value of observables and fidelity done in this study.