# Peer review of "Experimental QND measurements of complementarity on two-qubit states with IonQ and IBM Q quantum computers"

_SciPost Physics Core_

## Round 1 · Referee Report · Anonymous · 2021-6-4

Strengths

1.- Use of cloud quantum computing services in different platforms: superconducting qubits (IBM-Q) and Trapped ions (IonQ).

Weaknesses

1.- Preparation and characterization of Bell states in quantum hardware is a decade old.
2.- Poor motivation of the importance of quantities of interest.
3.- Goal of the paper is not met: the results superficially assess that one platform is slightly better than the other without any further details.
4.- Discussion about scalability of the platforms without any theory and/or figure to support it.

Report

In this manuscript the authors perform a quantum computation to characterize the quality of Bell states prepared on two different quantum hardware platforms via cloud services.
The authors calculate the concurrence, visibility and predictability of various quantum states by performing Quantum Non Demolition (QND) measurements using additional qubits and circuitry.
They use these quantities to assess the performance of the quantum hardware and make a comparison between the two platforms.

Unfortunately I am unable to recommend this work for publication in SciPost Core.
The main reason for this decision is that the manuscript does not meet the expectations of the journal;
Bell state preparation and characterization is more than a decade old and there is a vast literature about measuring, characterizing and studying Bell states in several platforms (see 10.1126/science.1130886 and Phys. Rev. A 81, 062325 (2010) as two of the first articles about it), thus not addressing a new or unsolved problem of the field.
Moreover, the results of this work are a basic quantum device benchmark step performed in any quantum computing lab, therefore the manuscript does not provide new results that help advancing the field or increase our understanding about quantum state characterization.
Finally, the conclusion is mostly focus on the scalability of the platforms without providing any data to back the claims in the form of computation times, latency in interaction with the cloud service, calibration measurements, etc.

  • validity: poor
  • significance: poor
  • originality: low
  • clarity: ok
  • formatting: acceptable
  • grammar: good

Author:  Christophe Galland  on 2021-06-08

(in reply to Report 1 on 2021-06-04)
Category:
answer to question
reply to objection

We thank the Referee for their report and their constructive criticism, to which we would like to reply below. We hope that we will be given the chance to submit a correspondingly revised manuscript.

  1. One of the main criticisms from Referee 1 is that "Preparation and characterization of Bell states in quantum hardware is a decade old". We agree with this comment, but find it surprising, since the novelty of our study lays in the non-demolition measurement of a general two-qubit state (including Bell states). In case we missed references performing QND measurements of entanglement, predictability and concurrence on a two-qubit system, we would be grateful if the Referee can point us to these papers.
    From the current report, we feel that the Referee missed the principal element of interest, that is the non-demolition measurement on a bipartite quantum system, performed in state-of-the-art commercial quantum computers. The development of non-demolition measurement schemes could lead to new algorithms, leveraging entanglement purification for example, in particular in the era of noisy intermediate scale quantum devices, and in quantum sensing. The study of complementarity relations, on the other hand, has ramifications in fundamental tests of quantum theory.

  2. We thank the Referee for mentioning that the measurement of the quantities should be better motivated. One motivation is the direct operational character of these quantities, which allows our results to be interpreted in terms of the usable coherences existing at the single and bipartite level in the system after the QND measurement. Moreover, the concept of complementarity is central to quantum mechanics, and it is currently being studied experimentally on similar platforms (see e.g. https://arxiv.org/abs/2105.07832). If given the opportunity, we plan to modify the manuscript accordingly, by motivating these measurements with concrete examples highlighting their relevance.

  3. While it is true that our study shows that one machine is performing better than the other, we do acknowledge that a detailed understanding of all the parameters accounting for differences between the platforms is beyond the scope of our paper. We are currently working precisely on this issue, in particular on the impact of the optimisation algorithm used in Qiskit on the results and its resilience to fluctuations in hardware parameters such as gate fidelities. Yet, with the present study, we already provide valuable information to the growing community of “quantum computer scientists” about the capabilities and limitations of currently available architectures. We think that our work can be particularly valuable to non-physicists who may not have a detailed understanding of the hardware limitations. Finally, echoing to point 2 above, we stress the potential value of our work in laying the ground for future fundamental tests of complementarity in increasingly complex multi-partite scenarios.

  4. We acknowledge that the discussion about scalability is lacking depth, and is not fully relevant where it appears in the conclusion. We propose to remove it from a future version.

We remain available to answer further comments or clarify our replies.

---

## Editorial Decision

editor-in-charge_assigned